# Potential Nociceptive Role of the Thoracolumbar Fascia: A Scope Review Involving In Vivo and Ex Vivo Studies

**DOI:** 10.3390/jcm10194342

**Published:** 2021-09-24

**Authors:** Larissa Sinhorim, Mayane dos Santos Amorim, Maria Eugênia Ortiz, Edsel Balduino Bittencourt, Gianluca Bianco, Fabiana Cristina da Silva, Verônica Vargas Horewicz, Robert Schleip, William R. Reed, Leidiane Mazzardo-Martins, Daniel F. Martins

**Affiliations:** 1Experimental Neuroscience Laboratory (LaNEx), University of Southern Santa Catarina, Palhoça 88137-272, Brazil; larissasinhorim@hotmail.com (L.S.); mayane_amorim@hotmail.com (M.d.S.A.); gegephisio@gmail.com (M.E.O.); edselb64@gmail.com (E.B.B.); gianluca.bianco@posturologiaclinica.eu (G.B.); vhorewicz@hotmail.com (V.V.H.); danielmartinsfisio@hotmail.com (D.F.M.); 2Postgraduate Program in Health Sciences, University of Southern Santa Catarina, Palhoça 88137-272, Brazil; 3Human Movement Sciences Graduate Program, College of Health and Sport Science at Santa Catarina State University, Florianópolis 88080-350, Brazil; 4Coastal Health Institute, Jacksonville, FL 32224, USA; 5Research Laboratory of Posturology and Neuromodulation RELPON, Department of Human Neuroscience, Sapienza University, 00147 Rome, Italy; 6Istituto di Formazione in Agopuntura e Neuromodulazione IFAN, 00147 Roma, Italy; 7Cirklo Health Education, Porto Alegre 90150-003, Brazil; fabisis@gmail.com; 8Department of Sport and Health Sciences, Technical University of Munich, 80799 Munich, Germany; 9Department for Medical Professions, DIPLOMA University of Applied Sciences, 37242 Bad Sooden-Allendorf, Germany; 10Department of Physical Therapy, University of Alabama at Birmingham, Birmingham, AL 35294, USA; wreed@uab.edu; 11Rehabilitation Science Program, Departments of Physical and Occupational Therapy, School of Health Professions, University of Alabama at Birmingham, Birmingham, AL 35294, USA; 12Postgraduate Program in Neuroscience, Center of Biological Sciences, Federal University of Santa Catarina, Florianópolis 88040-900, Brazil; leidiane.mazzardo@ufsc.br

**Keywords:** fascia, in vivo, ex vivo, innervation, pain, thoracolumbar fascia, nociceptor, lower back pain, scoping review

## Abstract

Nociceptive innervation of the thoracolumbar fascia (TLF) has been investigated over the past few decades; however, these studies have not been compiled or collectively appraised. The purpose of this scoping review was to assess current knowledge regarding nociceptive innervation of the TLF to better inform future mechanistic and clinical TLF research targeting lower back pain (LBP) treatment. PubMed, ScienceDirect, Cochrane, and Embase databases were searched in January 2021 using relevant descriptors encompassing fascia and pain. Eligible studies satisfied the following: (a) published in English; (b) preclinical and clinical (in vivo and ex vivo) studies; (c) original data; (d) included quantification of at least one TLF nociceptive component. Two-phase screening procedures were conducted by a pair of independent reviewers, after which data were extracted and summarized from eligible studies. The search resulted in 257 articles of which 10 met the inclusion criteria. Studies showed histological evidence of nociceptive nerve fibers terminating in lower back fascia, suggesting a TLF contribution to LBP. Noxious chemical injection or electrical stimulation into fascia resulted in longer pain duration and higher pain intensities than injections into subcutaneous tissue or muscle. Pre-clinical and clinical research provides histological and functional evidence of nociceptive innervation of TLF. Additional knowledge of fascial neurological components could impact LBP treatment.

## 1. Introduction

The process by which intense thermal, mechanical or chemical stimuli are detected by a subpopulation of nociceptive peripheral nerve endings is called nociception [1,2,3,4,5]. Nociceptors have a peripheral and central axonal branch that innervates their target organ and spinal cord, respectively [1,2,6,7]. Their cell bodies are located in the dorsal root ganglia (DRG) for the trunk and extremities and in the trigeminal ganglion for the face [8,9,10]. There are two major classes of nociceptors [1,2,3,6,7]. The first includes medium diameter myelinated (Aδ) afferents that mediate acute, well localized nociception [2,3,4,6,7]. The second class of nociceptor includes small diameter unmyelinated “C” fibers that convey poorly localized nociception. Most C fibers are polymodal (heat and mechanical sensitive) [2,3,4,6,7]. Often, heat-responsive unmyelinated afferents develop mechanical sensitivity only in the injury setting [11]. These afferents are more responsive to chemical stimuli (capsaicin or histamine) and are likely to come into play when the chemical milieu of inflammation changes their properties [1,2,12,13,14]. However, not all C fibers are nociceptors. Some respond to innocuous cooling, while others respond to innocuous stroking of the hairy skin and appear to mediate pleasant touch [15].

C nociceptive fibers project to the superficial dorsal horn (laminae I & II) of the spinal cord, which is organized into anatomically and electrophysiological distinct laminae [1,2,6,7]. The heterogeneity of C fibers has been demonstrated by their neuroanatomical and molecular characterization [16]. For example, the so-called peptidergic population of C nociceptors release the neuropeptides, substance P, and calcitonin gene-related peptide (CGRP); the nonpeptidergic population of C nociceptors express the c-Ret neurotrophin receptor that is targeted by glial-derived neurotrophic factor (GDNF) [1,2,17]. It has been demonstrated that CGRP and SP can act synergistically to promote inflammation and nociceptor sensitization, since they are released by the peripheral or central projections of primary afferent neurons [18,19]. In the periphery, CGRP and SP induce the production and release of inflammatory cytokines, thus contributing to the peripheral sensitization of nociceptive neurons [20]. Centrally, when released into the spinal cord, CGRP and SP also induce central sensitization by acting on dorsal horn neurons [21]. Interestingly, it has been shown that CGRP can inhibit the degradation of SP and thus prolong its action [22]. Finally, CGRP may contribute to mechanical hyperalgesia or allodynia by enhancing the response of wide dynamic range neurons to cutaneous stimulation [23,24].

Musculoskeletal (MSK) pain is defined as acute or chronic pain that affects bones, muscles, ligaments, tendons, and even nerves [25,26,27,28,29,30]. According to the World Health Organization (WHO), 1.75 billion people have some form of chronic MSK pain [31]. Inadequately managed MSK pain can adversely affect quality of life and impose significant socioeconomic problems. For example, lower back pain (LBP) is the main contributor to disability worldwide [25,26,27,28,29,30]. However, the mechanisms that underlie its development remain poorly understood [26,27,28,29,30]. It has been demonstrated that there is nociceptive innervation in connective tissues associated with nerves [32], tendons [33], and joints [34,35,36,37,38]. Together, these findings strongly suggest a role in the etiology of pain, and additional studies of nociceptive innervation of connective tissue will help to relate connective tissue dysfunction with LBP [39,40]. To date, it has been demonstrated that when healthy subjects underwent an injection of hypertonic saline into the thoracolumbar fascia (TLF), it resulted in pain of greater intensity, more unpleasant quality, and spreading over a greater area of the back and lower limbs, compared to similar injections into muscle or subcutaneous tissue [41,42,43,44]. Furthermore, this pain was associated with chemical stimulation rather than mechanical distention [42]. These studies suggest that the TLF makes a distinct contribution to nociception.

The TLF is a major connective tissue structure that can resist high tensile loads due to its fibers being oriented in multiple planes [45,46,47]. It has been shown that the thoracolumbar fascia shear strain was ~20% lower in human subjects with chronic lower back pain [48]. TLF consists of several layers. The posterior layer covers the deep muscles (deep lamina) of the back and attaches with the spinous processes via the supraspinous ligament. This posterior layer of the TLF consists of two separate laminae. The superficial lamina forms a continuation of the aponeurosis of the latissimus dorsi muscle as well as of the aponeurosis of the gluteus maximus muscle. It usually contains many dense collagen fibers crossing to the contralateral side below L2, which constitutes a very rare feature within mammalian anatomy and is probably associated with a contralateral force transmission between latissimus dorsi on one side and the gluteus maximus of the opposite side during human locomotion. This layer is also continuous with the external oblique musculature [49]. Anterior is the deep lamina, which blends with the supraspinous ligament cranially and with the long head of the biceps femoris muscle via the sacrotuberous ligament caudally [50,51]. The deep lamina is usually much thinner than the superficial lamina. Both laminae contain a multitude of smaller blood vessels, but do not contain any large arteries or veins [52,53]. Although the biomechanical properties of TLF are extensively documented and discussed in the literature, the neural properties have relatively received very little attention to date [39].

Despite nociceptive innervation of the TLF having been investigated over the past few decades in various animal models and humans using neuroanatomical and molecular techniques, these studies have not been compiled and their collective findings appraised. The purpose of this scoping review was to assess current scientific knowledge on the nociceptive innervation of the TLF in animal and human in vivo and ex vivo experiments to act as a resource for future clinical research targeting TLF in clinical treatment.

## 2. Materials and Methods

A scoping review methodology was selected to compile and evaluate data relating to the current state of scientific knowledge available in the literature related to TLF nociception and to identify the gaps that need to be addressed. The framework chosen was based on the guidelines provided by Arksey e O’Malley (2005) [54], Preferred Reporting Items for Systematic reviews, extensions of Meta-Analyzes, Scoping Reviews (Appendix A), and randomized clinical trials. Human and preclinical studies were included.

### 2.1. Step 1: Identifying the Research Question

The aim of this scoping review was to identify the role of nociceptive innervation of the TLF in musculoskeletal pain.

### 2.2. Step 2: Identifying Relevant Studies

A search strategy was developed using a combination of relevant keywords (Appendix A). The search was conducted in June 2021 in the electronic databases of PubMed, ScienceDirect, Cochrane, and Embase. Based on the scoping review methodology, the search filters used were the following: (a) pre-clinical studies; (b) clinical trials; (c) until the year 2021; (d) full text. Additional filters used were the following: (1) Species: human and animals; (2) Language: English; (3) Sex: female and male.

Both animal models and human studies were included. A combination of DeCs descriptors in English was used: fascia, pain, and lower back pain. After excluding duplicate articles, titles and abstracts were read for later selection to read the full text. Through the search for the DeCs descriptors, we found the following numbers of articles: (a) fascia and pain, 217 articles; (b) fascia and nociceptive pain, 13 articles (all duplicates); (b) fascia and nociceptive pain and innervation (in vivo studies/ex vivo studies), 15 articles (all duplicates); (e) fascia and nociceptive pain and myofascial pain syndrome, 3 articles (a single article duplicated); (f) fascia and nociceptive pain and lower back pain, 9 articles (all duplicates).

### 2.3. Step 3: Study Selection


Inclusion and Exclusion Criteria


The inclusion criteria consisted of the following: (a) studies published in English; (b) complete studies (c) preclinical and clinical (in vivo and ex vivo) studies; (d) original data studies; (e) studies involving the relationship between fascia and pain; (f) included quantification of at least one analysis of any nociceptive component of thoracolumbar fascia.

Studies were excluded if classified as the following: studies involving the relationship between fascia and proprioception, usage protocols, clinical intervention, fasciotomy, surgical procedures and medicine, practical guidelines, unpublished manuscripts, dissertations, reviews, expert comments, books and/or chapters of books, registered reports, conference proceedings.

Records screened by title and abstracts numbered 214. After reading the abstracts, 154 were excluded due to failure to meet the inclusion criteria. Fully read articles numbered 60, and of these, 48 were excluded because the content did not address the proposed theme, included interventions, or were related to fasciotomy, surgical procedures associated with the usage of medication, crural fascial, or proprioceptive innervation of facial tissue (Figure 1). In the end, two tables containing a total of 12 articles were developed: (a) Table 1—ex vivo studies and (b) Table 2—in vivo studies.

### 2.4. Screening and Agreement

The research articles were reviewed and selected in two phases. Phase I consisted of screening titles and abstracts to include possible relevant studies and to exclude irrelevant studies. Phase II consisted of screening full texts of studies previously identified as possibly relevant to select eligible studies. Screenings in phases I and II were performed by two independent reviewers (L.S., M.A.) and any discrepancy in relation to the study’s eligibility was mediated by a third reviewer (D.F.M., *n* = 2).

### 2.5. Step 4: Data Charting

The following data were extracted from eligible studies: (a) references; (b) animals, disease model; (c) method; (d) main results (Table 1 and Table 2). The extraction parameters were defined jointly by two authors. Data extraction was performed by one author and verified by a second author to minimize any discrepancy.

### 2.6. Step 5: Collating, Summarizing, and Reporting the Results

The data were summarized descriptively according to the following items:i.Preclinical: including in vivo and ex vivo studies related to nociceptive investigation of TLF.ii.Clinical: including in vivo and ex vivo studies related to nociceptive investigation of TLF.

## 3. Results

### 3.1. Basic Numerical Analysis

The search of the database carried out on 30 June 2021 resulted in 258 articles. After the removal of duplicates, 214 articles had their titles and abstracts screened in Phase I, and those articles considered to be eligible for Phase II were screened. Fifty-eight articles were selected to be read in full, and of these, 48 were excluded due to failure to meet eligibility requirements. Thus, a total of 12 articles were included in this review, all of which were published prior to the year 2021. These 12 articles were divided into two groups: (1) ex vivo studies and (2) in vivo studies.

### 3.2. Ex Vivo Studies

A total of seven studies evaluated and quantified the nociceptive nerve fibers terminating within the nonspecialized connective tissues in the lower back. Specific immunohistochemistry assays included immunofluorescence (5/7), calcitonin gene-related peptide (CGRP) immunoreactivity (5/7), substance P (SP) immunoreactivity (4/7), and retrograde labeling (2/7). Species selected for ex vivo analysis included mice (2/5), rats (4/7), and humans (3/7). Anatomical structures analyzed included the TLF (7/7), lower back muscles (7/7), and DRG (2/7). Detailed information regarding the immunohistochemistry assay, species, and outcome measures utilized can be found in Table 1.

### 3.3. Nociceptive Innervation of TLF: Histological Evidence

A total of seven studies characterized the distribution of the nociceptive nerve fibers terminating within the TLF [39,50,55,56,57,58,59]. All these studies investigated whether fasciae associated with muscle are also innervated by peptidergic nociceptive sensory fibers. Collectively, these studies regarding “naïve” fascia (mice, rats, and human) reported the following: (a) 60–88% DRG cells with terminations within the collagen matrix of connective tissue in the lower back also expressed calcitonin gene-related peptide (CGRP) immunoreactivity in rats [55]; (b) The TLF contained a higher density of sensory nerves, with three times more CGRP-immunoreactive fibers compared to back muscles (latissimus dorsi); however, around half of these lacked SP-immunoreactivity in mice [56], (c) CGRP-immunopositive fibers in the outer and inner layer of TLF and SP-containing nerve fibers existed exclusively in the outer layer of TLF in rats [57]; (d) in rats, fibers immunoreactive to protein gene product (PGP) 9.5, tyrosine hydroxylase (TH), CGRP, and SP were found in the subcutaneous tissue and the superficial lamina of the posterior TLF layer, and in humans, the majority of peptidergic nerve endings were located in the subcutaneous tissue and in structures comparable with superficial lamina and deep lamina of the posterior rat TLF layer [50]. Two studies investigated nociceptive-related responses to experimental irritation of the TLF, which resulted in the following: (a) in rats with CFA injections made in the spinous processes L4 and L5, the CGRP-immunopositive fiber length increased significantly only in the inner layer of TLF and SP-positive structures were found in the deep lamina of the posterior TLF layer [57]; (b) in rats, inflammation was induced by electrically stimulating the dorsal roots L3–L6 at an intensity supramaximal for unmyelinated fibers. Specifically, SP and CGRP are known to cause plasma extravasation by increasing the permeability of capillaries close to the endings. Neurogenic plasma extravasation in the form of dark patches of Evans blue in the TLF were observed [57].

### 3.4. In Vivo Studies

A total of five studies evaluated and quantified nociceptive outcomes after electrical and chemical stimulation of the TLF relating to its contribution to LBP [42,43,44,49,50]. Of these five studies, two investigated the role of chemical (3/5) and electrical (2/5) irritation of the TLF on measures related to nociceptive innervation of the lower back as a potential source of LBP. Specific outcomes included electrophysiological recordings from dorsal horn neurons in spinal (2/5), pressure pain threshold (algometer) (2/5), pain distribution (1/5), pain quality (1/5), and electrical detection threshold (1/5). Two studies made systematic recordings from dorsal horn neurons in the spinal cord to obtain information about the spinal nociceptive processing from the TLF in naïve rats. Species selected for in vivo analysis included rats (2/5) and humans (3/5). Anatomical site of analysis included the TLF (5/5), lower back muscles (5/5), and dorsal horn neurons in spinal (2/5). Detailed information regarding the species and outcome measures can be found in Table 2.

### 3.5. Nociceptive Innervation of TLF: Functional Evidence

A total of five studies evaluated the role the nociceptive input through electrical and chemical stimulation of the TLF in relation to LBP [42,43,44,60,61]. Of these five studies, three investigated the role of chemical irritation of the TLF on measures related to nociceptive innervation of the lower back as a potential source of LBP. These studies regarding chemical irritation of fascia (in rats and humans) reported the following: (a) in rats with bilateral complete Freund’s adjuvant (CFA) [60] with injections made in the spinous processes L4 and L5 or intrafascial (L3) [61], there was a significant increase from 4% to 15% in the proportion of neurons responsive to input from the TLF and neurons in the L3 spinal segment that did not receive input from the fascia in control animals; (b) healthy volunteers received ultrasound-guided bolus injections of hypertonic saline into the erector spinae muscle, in the deep lamina of the posterior TLF layer, and the overlying subcutis. Hypertonic saline injections into the TLF resulted in longer pain duration and higher pain intensities than injections into subcutaneous tissue or muscle. Pain radiation and pain affect evoked by fascia injections exceeded those of the muscle and the subcutaneous tissue. Pain descriptors (burning, throbbing, and stinging) suggested innervation by both A- and C-fiber nociceptors [42]. 

Two studies [43,44] used electrical stimulation with high-frequency pulses in the multifidus muscle and the overlying TLF through bipolar concentric needle electrodes placed at lumbar level (L3/L4) to assess distinct differences between nociceptive innervation of lower back fascia and muscles in humans. These studies reported the following: (a) lower electrical pain thresholds and higher pain ratings for fascia for both tissues. Fascia high-frequency stimulation increased fascia pain ratings 2.17 times compared to control; muscle high frequency stimulation (HFS) decreased pain sensitivity of the overlying fascia by 20%; potentiation by fascia HFS was similar to that of skin HFS, followed over 60 min post-HFS [43]; (b) Factor analysis of the sensory descriptors revealed that superficial thermal and superficial mechanical factors were more pronounced for fascia than muscle, whereas deep pain was more pronounced for muscle than fascia [44].

## 4. Discussion

The main objective of this scoping review was to assess current knowledge regarding nociceptive innervation of the TLF to better inform future mechanistic and clinical TLF research targeting LBP treatment. The results observed here, from a collective compilation and evaluation of studies on TLF nociceptive innervation, demonstrate that rodents and humans have an important nociceptive afferent TLF innervation and that the stimulation of this structure can lead to a central sensitization process in the long term. In this sense, these findings support the idea—through neurophysiological substrates related to nociceptive innervation—that TLF may be, at least in part, the origin of LBP. Although we recognize that the sensory innervation of the TLF encompasses proprioceptive, autonomic, and nociceptive components, here, we focus on nociception as a direct cause of pain. In that regard, articles for this review were restricted to preclinical and clinical including ex vivo (*n* = 7) and in vivo (*n* = 5) studies related to nociceptive investigation of TLF.

### 4.1. Ex Vivo Studies

Ex vivo studies were constituted primarily of specific immunohistochemistry assays to neuropeptides [39,50,55,56,57]. Mice [56], rats [39,50,55,56,57], and humans [50,57] presented positive immunoreactivity to SP and CGRP in the TLF, a nonspecialized connective tissue in the lower back [51]. Furthermore, functional evidence confirmed the hypothesis of nociceptive innervation of TLF, since the chemical irritation of the fascia by local injection of CFA significantly increased SP- and CGRP-immunopositive fiber in the deep lamina of the posterior TLF layer. In addition, electrical stimulation of the dorsal lumbar roots at an intensity for unmyelinated fibers increased the permeability of capillaries close to the endings, causing neurogenic plasma extravasation, a phenomenon known to be caused by SP and CGRP [57]. DRG neurons traced from the collagen matrix of connective tissue in the lower back support the finding that populations of CGRP-immunoreactive fibers innervate the TLF [55,56]. Innervation density was higher in the TLF compared to muscles of the back, supporting the view that the TLF may make an under-recognized contribution to chronic back pain. 

### 4.2. In Vivo Studies

In vivo and functional studies were constituted primarily of articles that evaluated nociceptive outcomes after chemical [42] and electrical [43,44] stimulation of the TLF relating to its contribution to LBP. In humans, the pain evoked by ultrasound-guided bolus injection of hypertonic saline [42] or electrical stimulation [43] in TLF tissue exceeded those of muscle. Furthermore, the pain after chemical stimulation of TLF was described by the patients in rather extreme terms such as cutting, tearing, and stinging, which suggests innervation by both A- and C-fiber nociceptors [42,62,63]. In contrast, muscle-derived pain was predominantly attributed to sensory qualities like throbbing and pounding [44]. It is understood that C- and A-cutaneous afferents correlate with group IV and III muscle afferents [62]. However, the fact that muscle-associated pain differs from cutaneous-associated pain strongly suggests that the neurophysiological mechanisms are not similar. Studies have suggested the term fasciatome for segmental innervation deep fascia which helps to differentiate between pain of cutaneous origin and pain of muscular origin [50,51,64]. Unlike the dermatome, which represents the portion of tissue composed of skin, hypodermis, and superficial fascia supplied by all cutaneous branches of an individual spinal nerve, the fasciatome is the portion of the deep fascia supplied by the same nerve root and organized along lines of force to emphasize the main directions of movement. When the dermatome is altered, it signals the location of pain, while the fasciatome demonstrates pain irradiation through the organization of the fascial anatomy [50,51,64]. This is evident when electrical stimulation of the muscle nerves produces only a single pain, as opposed to electrical stimulation of the cutaneous nerves which presents pain in two phases [63]. Corroborating the findings of our scoping review, it has been previously described that fascial pain quality is different, as cutaneous pain is well localized and described as stabbing, burning, or cutting, while muscle pain tends to be referred to and described as tearing, cramping, or pressing [63]. Finally, electrophysiological recordings from dorsal horn neurons in spinal cord rats showed that the nociception evoked by the injection of a chemical irritant in the lumbar spinous process increased the proportion of neurons responsive to input from TLF and neurons in the L3 spinal segment that previously did not respond to input from the fascia in naive animals [60]. Interestingly, this study also showed that most dorsal horn neurons that receive nociceptive input from the TLF are convergent and have additional input from other tissues, such as skin and lumbar muscles, at least in rats [60]. The nociceptors located in the fascia are possibly the beginning of the nociceptive pathway from the lumbar region soft tissue to the spinal dorsal horn and to other central locations from there. Since input from all soft tissues of the lower back converge onto the same dorsal horn neurons, no separate pathway from fascia to higher centers appears to exist. This may be the explanation for why a pain from the TLF is difficult to distinguish from the pain from other soft tissues of the lower back [60,61]. Hoheisel et al. (2015) [61] made an intrafascial CFA injection in the TLF and registered dorsal horn neuron activity in the lumbar segment L3. They observed that the proportion of neurons with input from all deep somatic tissues rose from 10.8% to 33.3% when compared with control. One of the most relevant findings in animals with fasciitis was the appearance of new receptive fields in deep somatic tissues beyond the fascia, located outside the lumbar region and extending to the distal hindlimb. The appearance of new receptive fields could be a possible explanation for the spread of pain in patients with LBP.

### 4.3. Limitations

Limitations associated with this scoping review include selecting publications in English only and free full text publications, which could potentially reduce the number of studies being retrieved from the literature search.

### 4.4. Clinical Perspectives and Future Directions

Published guidelines have proposed non-pharmacological approaches such as exercise and physical therapy (massage, acupuncture, spinal manipulative therapy) as first-line treatments for lower back pain primarily due to concerns about the risk–benefit ratio of opioids and suboptimal opioid-related results in clinical trials [65,66,67]. A major gap in LBP research includes the contribution of musculoskeletal tissues (unspecialized muscles and connective tissues) in the lower back of the spine [68]. Although not well understood, the pathophysiology of non-specialized connective tissues, their relationship to LBP, and other conditions of chronic musculoskeletal pain can include fibrosis, chronic inflammation and neuroplastic changes, such as altered sensitivity [69,70,71,72,73]. Thus, the results of the present study highlight the role of connective tissue innervation in the development of lower back pain by demonstrating the presence of sensory nerve fiber endings within the collagen matrix of non-specialized connective tissue, and that structural innervation of these connective tissues is important for functional changes associated with neurogenic inflammation and persistent pain. Based on the results of this review, it is suggested that the treatment of TLF should also be considered in LBP therapeutic approaches.

Despite the recent increase in the number of basic scientific investigations describing the nociceptive innervation of TLF, this review has identified numerous areas that require further study to advance the field and deepen our understanding of the fascia approach. Although peripheral evidence exists and provides strong evidence for neurophysiological substrates of the nociceptive innervation of the fascia, there is limited evidence related to the central (supra-spinal) mechanisms involved. Future preclinical studies related to techniques aimed towards treatment (i.e., manipulation/mobilization) of TLF mimicking the clinical setting could be useful to advance knowledge and help with the treatment of musculoskeletal pain. In this sense, preclinical studies that record the results immediately after (or shortly thereafter) the delivery of fascial manipulation, long-term or longitudinal studies, and studies that investigate the physiological impact of different fascial manipulation dosage are needed. As in the study of França et al., 2020 [74], preclinical studies are just beginning to recognize and demonstrate that analgesic modulation related to fascial manipulation involves complex mechanistic interactions. Thus, preclinical studies that investigate endogenous neurological systems such as endogenous opioids [75], endocannabinoids [76], adenosinergic system [77], and/or neuroimmune contributions [74,78] are needed, and these effects may require that certain fascial manipulation dosage thresholds be achieved. Preclinical studies investigating the effects of fascial manipulation on gene expression, neurotransmitter/neuropeptide/cytokine release, mechanosensitive ion channel activation, neuroimmune response, global cortical/spinal circuit connectivity, and descending inhibition [77,78] are needed to advance the manual therapy field and deepen our understanding of fascial manipulation.

## 5. Conclusions

While more studies are clearly needed, findings from this review based on histological, immunohistochemical, electrophysiological, behavioral, and clinical evidence support the view that the TLF potentially plays a major role in clinical LBP, or at least makes significant contribution.

## Figures and Tables

**Figure 1 jcm-10-04342-f001:**
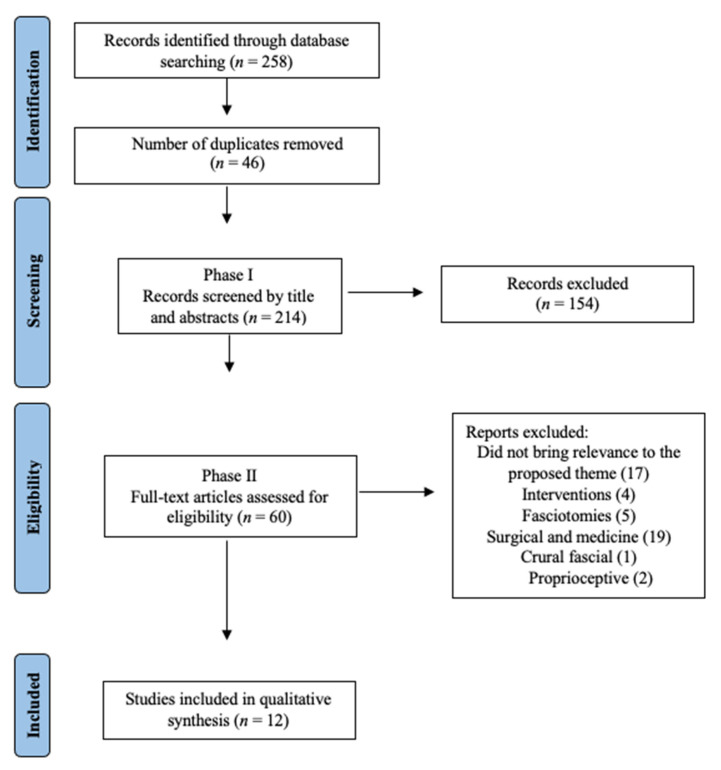
Flowchart diagram (file in attachment).

**Table 1 jcm-10-04342-t001:** Ex vivo studies.

Studies (Authors, Years)	Specie, Condition, *n*	Method	Main Findings
Corey et al., 2011 [55]	Rats, Naïve, *n* = 17	Three-dimensional reconstructions of thick tissue sections (lower back L1–L6) Fast Blue retrograde labeling (lower back muscle—DRG) and immunofluorescence with CGRP antibody	60–88% DRG cells with terminations within the collagen matrix of connective tissue in the lowr back also expressed CGRP.
Barry et al., 2015 [56]	C57Bl/6 mouse, Naïve, 4–8 per group	Immunohistochemistry: TLF were multiple-labeled using antibodies to the pan-neuronal marker neuron-specific enolase (NSE), and CGRP and SP Multiple-labeling immunofluorescence and retrograde axonal tracing	The TLF contained the same proportions of nerve fiber subpopulations found in back muscles. The TLF contained a higher density of sensory nerves, with three times more CGRP-immunoreactive fibers compared to back muscles (latissimus dorsi). However, around half of these lack SP-immunoreactivity.
Mense et al., 2016 [57]	Rats, Naïve, *n* = 5 Fascia inflammation—CFA, *n* = 5 Dorsal root electrical stimulation, *n* = 4	Immunohistochemical antibodies to CGRP, SP, and TRPV1 Functional data Neurogenic plasma extravasation (Evans Blue)	In naïve rats, CGRP-immunopositive fibers were found in the outer and inner layer of TLF. SP-containing nerve fibers existed exclusively in the outer layer of TLF. In inflamed (CFA) rats, The CGRP-immunopositive fiber length increased significantly only in the deep lamina of the posterior TLF layer SP-positive structures were found in the same layer of TLF. In inflamed (electrical stimulation) rats, Neurogenic plasma extravasation existed in the form of dark patches of Evans blue in the TLF
Tesarz et al., 2011 [50]	Rats, Naïve, *n* = 8 Specimens Humans TLF *n* = 3	Immunohistochemical antibodies to PGP 9.5, TH, CGRP, and SP	In rats, Fibers immunoreactive to PGP 9.5, TH, CGRP, and SP were found in the layer adjacent to the subcutaneous tissue and the superficial lamina of the posterior TLF layer. In humans, The majority of peptidergic nerve endings were located in the subcutaneous tissue and in structures comparable with superficial and deep lamina of the posterior rat TLF layer.
Mense et al., 2019 [39]	Rats, Naïve and Fascia inflammation—CFA, *n* = NI Specimens Humans TLF *n* = NI	Immunohistochemical antibodies to PGP 9.5, TH, CGRP, and SP	In rats, Fibers immunoreactive to PGP 9.5, TH, CGRP, and SP were found in the TLF. The inflamed fascia exhibited a higher density of CGRP-immunoreactive and SP- immunoreactive units. In humans, Fibers immunoreactive to CGRP were the most frequent ones.
Marpalli et al., 2021 [58]	Specimens Humans TLF *n* = 20	Microscopy—Morphological morphometric study	The quantification of peripheral nerves and nociceptors were quantifiable in the deep lamina of TLF. The thickness of TLF and number of nerve endings in the sacral level was increased compared to that of thoracic vertebral levels.
Fede et al., 2021 [59]	C57Bl/6 mouse, Naïve, *n* = 7	Immunohistochemical: TLF were multiple-labeled using antibodies to Tyrosine Hydroxylase, S100, and PGP Transmission electron microscopy	The TLF presented free nerve endings and autonomic innervation.

Note: CFA—complete Freund’s adjuvant; CGRP—calcitonin gene-related peptide; DRG—dorsal root ganglia; SP—substance P; TLF—thoracolumbar fascia; TRPV1—transient receptor potential vanilloid 1—TH: tyrosine hydroxylase; NI—not informed.

**Table 2 jcm-10-04342-t002:** In vivo studies.

Studies (Authors, Years)	Species, Condition, *n*	Method	Main Findings
Schilder et al., 2014 [42]	Healthy humans; Ultrasound-guided bolus injections of hypertonic saline into the erector spinae muscle, the thoracolumbar fascia (TLF, posterior layer), and the overlying subcutis; *n* = 6 women, *n* = 6 men	Pressure pain threshold (algometer) Pain distribution (two-dimensional paper form body image) Pain quality (Pain Perception Scale, ‘‘Schmerzempfindungs-Skala’’ [SES])	Hypertonic saline injected into the fascia resulted in longer pain duration and higher pain intensities than injections into subcutaneous tissue or muscle. Pain radiation and pain affect evoked by fascia injection exceeded those of the muscle and the subcutaneous tissue. Pain descriptors (burning, throbbing, and stinging) suggested innervation by both A- and C-fiber nociceptors.
Schilder et al., 2016 [43]	Healthy humans; Electrical stimulation with high-frequency pulses in the multifidus muscle and the overlying TLF through bipolar concentric needle electrodes placed at lumbar level (L3/L4); *n* = 8 women, *n* = 8 men	Electrical detection threshold and pain threshold Pain distribution (two-dimensional paper form body image) Pain rating during HFS	Fascia vs muscle high-frequency stimulation (HFS): Electrical pain thresholds were lower and pain ratings were higher for fascia. For both tissues, pain ratings increased significantly. Fascia HFS increased fascia pain ratings 2.17 times compared with control site. Muscle HFS decreased pain sensitivity of the overlying fascia by 20%. Potentiation by fascia HFS was similar to that of skin HFS, followed over 60 min post-HFS.
Schilder et al., 2018 [44]	Healthy humans, Electrical stimulation with high-frequency pulses in the multifidus muscle and the overlying TLF through bipolar concentric needle electrodes placed at lumbar level (L3/L4); *n* = 8 women, *n* = 8 men	Pain quality (Pain Perception Scale, ‘‘Schmerzempfindungs-Skala’’ [SES])	Factor analysis of the sensory descriptors revealed the following: Superficial thermal (“heat pain” identified by the items “burning”, “scalding”, and “hot”) and superficial mechanical factors (“sharp pain” identified by the items “cutting”, “tearing”, and “stinging”) were more pronounced for fascia than muscle. Deep pain (identified by the items “beating”, “throbbing”, and “pounding”) was more pronounced for muscle than fascia.
Hoheisel et al., 2011 [60]	Rats, Naïve, *n* = 12; Rats, Treated with CFA, *n* = 32	Systematic electrophysiological recordings were made from dorsal horn neurons in spinal (Th13–L5) to obtain information about the spinal nociceptive processing from the TLF.	In naïve rats, 6–14% of the neurons in the spinal segments Th13–L2 had nociceptive input from the TLF. No neurons responsive to input from the TLF were found in segments L3–L5. In inflamed (multifidus) rats, There was a significant increase from 4% to 15% in the proportion of neurons responsive to input from the TLF. Neurons in spinal segment L3 responded to fascia input in animals.
Hoheisel et al., 2015 [61]	Rats, Intrafascial CFA injection, *n* = 25	Extracellular recordings were made from dorsal horn neurons in the lumbar segment L3, identification of receptive fields and behavioral experiments were performed.	In inflamed (Intrafascial CFA injection) rats, 11.1% of neurons in the spinal segment L3 received input after TLF inflammation. Compared with control, the proportion of neurons having input from all deep somatic tissues rose from 10.8% to 33.3%. Moreover, many neurons acquired new deep receptive fields, most of which were located in the hindlimb. Although the pressure pain threshold of the inflamed rats did not change, they demonstrated a reduction in exploratory activity.

Note: CFA—complete Freund’s adjuvant; TLF—thoracolumbar fascia; HFS—high-frequency stimulation.

## Data Availability

No new data were created or analyzed in this study. Data sharing is not applicable to this article.

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
