# Peer review of "Potential Nociceptive Role of the Thoracolumbar Fascia: A Scope Review Involving In Vivo and Ex Vivo Studies"

_jcm, 2021, doi:10.3390/jcm10194342_

Round 1
Reviewer 1 Report
This review is very interesting and well written and it permits a new insight about pain generation and nociception with a possible future impact for LBP treatment.
However, although the Authors specify that the literature revision is until January 2021, I think that in this year new important findings have been published about TLF and innervation. Two examples:
- "Evidence of a new hidden neural network into deep fasciae", Fede et al., 2021
- The morphological and microscopical characteristics of posterior layer of human thoracolumbar fascia; A potential source of low back pain. Marpalli et al., 2021
So, in my opinion, the review needs to be updated and revised before publication.
Furthermore, one other minor comment: I suggest to highlight on the text (in the introduction and aim) that this review is focalized on the nociceptive investigation of TLF, and not on other fasciae of the body.
Other comments:
- introduction: the description of C nociceptive fibers is too long and disconnected from the rest of the introduction
- introduction , pge 3, line 117: The epimysial fascia of the EO was in direct continuity with the posterior layer of TLF, Fan et al 2018
- introduction: Authors can add and discuss the paper of Langevin about LBP and reduced thoracolumbar fascia shear strain (Langevin 2011)
- Methods. supplementary data sheet 2 is not available
- flow chart: it could be useful to insert the inclusion criteria
- Tables: I suggest to divide tables in clinical and preclinical studies, not in in vivo and ex vivo, and maybe paragraphs 3.3 and 3.5 can be mixed together
- line 295: ?
- lines 347-348: please insert numbers
- lines 371-372: Authors can discuss the differences between fasciatome and dermatome. In fact, if altered, the dermatome shows clearly localized pain and the fasciatome irradiating pain according to the organization of the fascial anatomy
- limitations: publications in English and free full text publications.
Author Response
See enclosed Word document

Reviewer 2 Report
I am grateful for the opportunity to read a review on LBP with regards to the TLF. It was an interesting read despite the very limited number of papers on this particular topic. This paper does a very good job of directing the reader through the article review process and summarizes the published material well. I would highly recommend that the authors spend a bit more time in the introduction making sure the reader is up to date with the terminology that they will be reading in the summary tables. For example, I would recommend that the authors provide additional information on neuropeptides, substance P and specifically CGRP. Lines 350-359 might serve the reader better if they are moved to the introduction. These are indicated in “Main Findings” in the articles reviewed but the reader is not provided with the biological consequences of these molecules in the introduction. Why is CGRP being tracked in TLF research? What does it do in the connective tissue? How to they relate to C nociceptors? It would be nice to have more information on neuropeptides, substance P and specifically CGRP in the introduction so that the summary tables are more informative to the reader. Thank you for your consideration.
Also in the introduction, line 107, it is important to note that specialized connective tissue does NOT include nerve. Nerve is its own tissue type – of which there are only 4 in the body (epithelium, connective tissue, muscle and nerve). Please remove nerve from the list. I would also be more specific with the parts of joints, because joints themselves are not specialized connective tissue. The bones, articular cartilage, menisci and joint capsule are all derivatives of connective tissue. Thank you for your consideration.
Lastly, in table 1 the layers of the TLF are described as “inner” and “outer” while in the introduction only the words posterior and anterior are used to describe the TLF. Can we make sure that the vocabulary is consistent? Lines 119, 126, 242 main findings, 261, 267, 305, 342. Thank you for your consideration.
For the authors/editors, please carry over the Table Headings to the next page so that the column titles stay labeled for the reader. It is frustrating to flip the pages back and forth to read the titles associated with each table column. This is recommended for both Tables 1 and 2. Thank you for your consideration.
Tiny Edits
Line 40 “(d) included quantification” could be simply “inclusive”
Line 48 “Greater” to “Additional”
Line 101 end sentence after nerve. Remove “in particular” and change to “for example”. Move that sentence after problems on line 105. This keeps everything with MSK together and then transitions smoothly to LBP.
Line 118 period after layers. Delete of which. Start new sentence with The line 119
Line 126 what does “it” refer to?
Line 127 nuchal ligament is a continuation of the supraspinous ligament, not a separate structure. Simply a structural rename in the cervical region.
Line 151 the word sensory caught me because we have been using nociception in the review. Perhaps use sensory in the introduction so readers link the two
Line 162 I would remove the first sentence. It has already been stated.
Lines198, 206, 212 Are the section number correct? At first glance they seem incorrect.
Line 349 change to TLF
Line 419 considered
Line 430-431 confusing as written “of various of”?
Author Response
See enclosed Word document

Round 2
Reviewer 1 Report
I thank the Authors: they met the requests of the Reviewer. I think that this version of the manuscript can be accepted for publication.